# Expression Pattern and Functional Analysis of *MebHLH149* Gene in Response to Cassava Bacterial Blight

**DOI:** 10.3390/plants13172422

**Published:** 2024-08-30

**Authors:** Min Cui, Feifei An, Songbi Chen, Xindao Qin

**Affiliations:** 1Key Laboratory of Green Prevention and Control of Tropical Plant Diseases and Pests, Ministry of Education, School of Tropical Agriculture and Forestry, Hainan University, Haikou 570228, China; cm1998mc@163.com; 2Tropical Crops Genetic Resources Institute, Chinese Academy of Tropical Agricultural Sciences, Haikou 571101, China; aff85110@163.com

**Keywords:** cassava, *MebHLH149*, *Xpm* CHN01, plant hormones, virus-induced gene silencing, overexpression, protein–protein interaction

## Abstract

The significant reduction in cassava (*Manihot esculenta* Crantz) yields attributed to cassava bacterial blight (CBB) constitutes an urgent matter demanding prompt attention. The current study centered on the MebHLH149 transcription factor, which is acknowledged to be reactive to CBB and exhibits augmented expression levels, as indicated by laboratory transcriptome data. Our exploration, encompassing *Xanthomonas phaseoli* pv. *manihotis* strain CHN01 (*Xpm* CHN01) and hormone stress, disclosed that the *MebHLH149* gene interacts with the pathogen at the early stage of infection. Furthermore, the *MebHLH149* gene has been discovered to be responsive to the plant hormones abscisic acid (ABA), methyl jasmonate (MeJA), and salicylic acid (SA), intimating a potential role in the signaling pathways mediated by these hormones. An analysis of the protein’s subcellular localization suggested that MebHLH149 is predominantly located within the nucleus. Through virus-induced gene silencing (VIGS) in cassava, we discovered that *MebHLH149*-silenced plants manifested higher disease susceptibility, less ROS accumulation, and significantly larger leaf spot areas compared to control plants. The proteins MePRE5 and MePRE6, which are predicted to interact with MebHLH149, demonstrated complementary downregulation and upregulation patterns in response to silencing and overexpression of the *MebHLH149* gene. This implies a potential interaction between MebHLH149 and these proteins. Both *MePRE5* and *MePRE6* genes are involved in the initial immune response to CBB. Notably, MebHLH149 was identified as a protein that physically interacts with MePRE5 and MePRE6. Based on these findings, it is hypothesized that the *MebHLH149* gene likely functions as a positive regulator in the defense mechanisms of cassava against CBB.

## 1. Introduction

Cassava (*Manihot esculenta* Crantz), known as the “king of starch”, is the fourth most significant source of carbohydrates for human consumption in tropical regions, behind rice, sugar, and maize [1]. This versatile crop is a vital provider of sustenance and economic livelihood, supporting approximately 1 billion people worldwide. However, the threat posed by cassava bacterial blight (CBB), is caused by *Xanthomonas phaseoli* pv. *manihotis* (*Xpm*), poses a significant challenge in tropical regions, where cassava cultivation is widespread [2,3]. CBB is a devastating foliar and vascular disease that can lead to substantial yield losses, severely impacting both the economic viability of cassava farming and food security in these areas [4]. The management of CBB predominantly relies on the resistance inherent in the host plant. However, the emergence of new strains of *Xpm* that can overcome this resistance calls for continuous research and development efforts to identify and cultivate cassava varieties that are resistant to these strains [5]. In extreme cases, it can cause a significant decrease in cassava production, ranging from 50 to 75%, potentially leading to famine [6]. Consequently, enhancing the natural resistance of cassava to CBB is crucial to mitigate yield losses. Notably, Zhu et al. successfully isolated the strain *Xanthomonas phaseoli* pv. *manihotis* strain CHN01 (*Xpm* CHN01) using PacBio single-molecule real-time (SMRT) sequencing technology. This pathogen was found to have a genome consisting of a roughly 4.79 Mb circular chromosome and four plasmids. This discovery opens up new avenues for developing prevention and control strategies against CBB [7].

Since plant disease resistance requires the coordination of multiple defense response genes, transcription factors (TFs), as key regulatory factors of gene expression, play crucial roles in plant disease resistance processes. There are reports dating back to early times indicating that basic helix–loop–helix (bHLH) transcription factors can interact with resistance genes to regulate the synthesis of compounds such as flavonoids. As early as 1989, Lc, a product of the *R* gene, was the first plant protein reported to have a bHLH domain, involved in controlling the biosynthesis of flavonoids and anthocyanins in maize [8]. The bHLH transcription factors consist of two α-helices that mediate dimerization and a basic domain that binds to E-box DNA sequences [9]. bHLHs are important in promoting plant tolerance or adaptation to adverse environmental conditions [10]. bHLHs often cooperate with members of other transcription factor families to control the regulation and induction of the biosynthesis of various secondary metabolites, mainly including terpenes and anthocyanins, which play crucial roles in mediating the interaction between plants and their environment [11,12].

Additionally, it has been reported that many bHLH transcription factors, such as those involved in abscisic acid (ABA) and auxin signal transduction, participate in the regulation of various hormone signaling pathways, and many have been shown to participate in the jasmonic acid (JA) signaling pathway [13]. Cao et al. identified a tomato bHLH transcription factor, *SlJIG*, as a MeJA-inducible gene, which exhibits strong induction upon MeJA treatment. Knocking out *SlJIG* also leads to reduced expression of JA-responsive defense genes, indicating that *SlJIG* is a direct target of MYC2, forming the MYC2-*SlJIG* module, plays a role in terpene biosynthesis and resistance against cotton bollworm and *Botrytis cinerea* [14]. Yoodee et al. showed that pre-treatment with salicylic acid (SA) or methyl jasmonate (MeJA) 24 h before *Xpm* inoculation enhanced the defense responses of two cassava cultivars. The severity of the disease decreased by 10% in the SA/MeJA-induced resistant cultivar (HB60) and significantly by 21% in the susceptible cultivar (HN) [15]. Cao et al. characterized the biological function of Crf1, a bHLH transcription factor, in the development and pathogenicity of *P. oryzae* through functional genetics, molecular, and biochemical methods [16]. Members of the bHLH transcription factor family, including MYC3 and MYC4 from the same family as MYC2, interact with JAZ proteins, contributing to important functions such as growth inhibition, promotion of leaf senescence, and defense against diseases and pests [17,18,19,20]. Susceptibility of *TabHLH060*-overexpressing *Arabidopsis* plants to *Pseudomonas syringae* DC3000 were significantly enhanced [21]. Transgenic rice plants overexpressing *OsbHLH034* exhibited a JA-hypersensitive phenotype and increased resistance against rice bacterial blight [22].

However, the role of bHLH transcription factor family in CBB resistance in cassava is not fully understood. The is study aimed to analyze the expression pattern of the *MebHLH149* gene and its function in cassava response to CBB. This will provide a new perspective for future molecular breeding of disease-resistant cassava.

## 2. Results

### 2.1. Identification of Genes

In this research, we identified the MebHLH149 transcription factor, which response to CBB, by analysis of transcriptome data as reported in a prior study by An et al. [23].

The gene sequence under investigation was retrieved using the respective gene IDs on the online Phytozome platform. Phytozome v13 was the source for downloading sequences. The primers used in this study for all genes are compiled in Appendix A. Additionally, Appendix A contains the gene IDs for all genes examined in this study.

### 2.2. MebHLH149 Gene Is Involved in the Immune Response of Cassava

To explore the genes responding to cassava bacterial blight, RT-qPCR analysis indicated that among the *MebHLHs* genes, the expression levels of *MebHLH149* (Figure 1a) and related reactive oxygen species genes (Figure 1b) significantly increased after the pathogen *Xpm* CHN01 infected cassava. To investigate this phenomenon, RT-qPCR analysis demonstrated that the expression level of *MebHLH149* was still significantly upregulated in the early stage of pathogen *Xpm* CHN01 infecting cassava (Figure 1c).

The results suggested that the *MebHLH149* gene could interact with the pathogen in the early stage, and the *MebHLH149* gene was very likely to be involved in the immune response of cassava.

### 2.3. MebHLH149 Gene Expression Is Responsive to Plant Hormones ABA, MeJA, and SA

To explore the effects of these hormones on cassava, we treated the plants with foliar applications of MeJA, SA, and ABA, aiming to induce a stress response. The qPCR analysis indicated that the foliar applications of ABA (Figure 1d), MEJA (Figure 1e), and SA (Figure 1f) significantly induced the expression of the *MebHLH149* gene. Concurrently, we conducted real-time quantitative PCR to assess the expression of genes associated with the hormonal pathways: the *MeNCED6* gene, which is involved in the ABA pathway (Figure 1g), the *MeMYC2* gene, related to the MEJA signaling pathway (Figure 1h), and the *MeEDS1* gene related to SA pathway (Figure 1i). The results show that these genes exhibited pronounced upregulation in response to the respective hormone treatments.

Taken together, these results suggest that the *MebHLH149* gene in cassava is responsive to the plant the hormones ABA, MeJA, and SA.

### 2.4. Subcellular Localization of MebHLH149

The NovoPro online tool predicted and discovered that the amino acid sequence from the 92nd to the 131st of this protein was the nuclear localization signal sequence (Figure 2a). The resulting plasmid (pNC-Green-SubN-*MebHLH149*) expressed the MebHLH149 protein with a green fluorescent protein (GFP) tag at the N-terminus (Figure 2b). By fusing with the GFP protein and transiently expressing it in *Nicotiana benthamiana* leaves, we found that the fluorescence signal of MebHLH149-GFP was co-localized with the nuclei stained by DAPI in the *Nicotiana benthamiana* leaves. These results indicate that MebHLH149 is localized in the nucleus (Figure 2c).

### 2.5. Reduction in Cassava Resistance after MebHLH149 Silencing

On the 26th day after silencing (Figure 3a), RT-PCR analysis was performed on the collected tissue samples. The results showed that pCsCMV-*MebHLH149* had the target band at 750 bp (Appendix A). After the sequencing results were correct, the qPCR analysis indicated that the expression level of the *MebHLH149* gene was significantly downregulated compared to the pCsCMV-NC control group (Figure 3b). The results demonstrated that the transient transformation of the *MebHLH149* gene in cassava was successful.

Pathogen *Xpm* CHN01 was inoculated onto the new leaves of the silenced plants. The outcomes reveal significant disparity in the progression of CBB infection between the control and experimental groups. Specifically, cassava leaves in the control group, which received the empty vector pCsCMV-NC, showed minimal CBB infection. In contrast, *MebHLH149*-silenced plants presented leaves with conspicuously larger water-soaked spots (Figure 3c), which were three times larger compared to those of the control plants (Figure 3d). Additionally, we carried out a bacterial count analysis to evaluate the bacterial population dynamics. Initially, at 0 d and 1 d, there was no substantial differences in bacterial counts between the two groups of plants. However, by 2d, a notable increase in the bacterial count was witnessed, in the experimental group showing a significantly higher bacterial count than the control group (Figure 3e).

Collectively considering all the data, the reduction in cassava resistance after silencing of *MebHLH149* implies that this gene plays a positive regulatory role in the CBB resistance pathway.

### 2.6. MebHLH149 Regulates ROS Accumulation

After inoculating the pathogen *Xpm* CHN01 onto the plants with the *MebHLH149* gene silenced (0d, 1d, 2d), RT-qPCR analysis indicated that the expression of the *MebHLH149* gene in the experimental group was lower compared to the control group, suggests that this gene’s silencing inhibited its expression (Figure 4a).

Pathogens were inoculated onto the new leaves of silenced cassava, and the contents of reactive oxygen species were determined at 0d, 1d, and 2d. The results showed that the contents of hydrogen peroxide (Figure 4b) and superoxide anion (Figure 4c) in the experimental group were lower compared to the control group. Meanwhile, in the early stage of pathogen *Xpm* CHN01 infection, the expression levels of *MePOD12* and *MeGST23* genes related to reactive oxygen species were significantly upregulated. It is speculated that the *MePOD12* and *MeGST23* genes might be involved in the immune response to CBB in the early stage (Figure 4d). In contrast, at 0d, 1d, and 2d after the inoculation of pathogens following the silencing of the *MebHLH149* gene, the expression levels of *MePOD12* and *MeGST23* genes were significantly lower than those of the control group. As mentioned earlier, this downregulation was positively correlated with the content of ROS (Figure 4e,f).

The results imply that the silencing of the *MebHLH149* gene has an impact on ROS accumulation.

### 2.7. Overexpression of MebHLH149 Gene in Cassava

To explore whether transgenic plants overexpressing *MebHLH149* are resistant to CBB at later growth stages, we initially constructed an overexpression vector for the *MebHLH149* gene and introduced it into *Agrobacterium* LBA4404. Subsequently, we transformed the cassava friable embryogenic callus (FEC) using the *Agrobacterium* containing the recombinant plasmid pNC-Cam1304-MCS35S-*MebHLH149.* The outcomes reveal that the genetic transformation of the *MebHLH149* gene in cassava was accomplished successfully, yielding transgenic plant seedlings (Appendix A). Meanwhile, we found that the expression level of *MebHLH149* was relatively stable in the non-transgenic cassava variety SC8 as well as in other non-transgenic cassava varieties (SC9, ZME642). However, in transgenic plants, its expression level increased significantly, far exceeding the normal range, indicating initially that *MebHLH149* was overexpressed in the genetically transformed cassava (Figure 5).

### 2.8. MebHLH149 Regulates the Expression of MePRE5 and MePRE6

A protein–protein interaction network involving the MebHLH149 protein was established using the online STRING platform (Appendix A). Initially, we evaluated the influence of *MebHLH149* gene silencing on the expression levels of *MePRE5* and *MePRE6*. The results (Figure 6a) revealed a significant reduction in the relative expression levels of both *MePRE5* and *MePRE6* after the silencing of the *MebhLH149* gene, in contrast to the control plants transformed with (pCsCMV-NC). Contrary to the gene silencing of *MebHLH149*, the expression levels of *MePRE5* and *MePRE6* were significantly upregulated in the cassava with successful genetic transformation of *MebHLH149* (Figure 6b). Furthermore, as shown in Figure 6c, during the early stage of pathogen *Xpm* CHN01 infection in cassava, the expression levels of *MePRE5* and *MePRE6* showed a significant upward trend.

The above results suggest that *MePRE5* and *MePRE6* may be involved in the immune response to pathogen *Xpm* CHN01 and play a role in regulating their expression, indicating that there may be interactions among these proteins.

### 2.9. MebHLH149 Physically Interacts with MePRE5 and MePRE6

The results of the self-activation verification indicated that the proteins MebHLH149, MePRE5, and MePRE6 did not show self-dimerization, which was a prerequisite for further experimental validation (Figure 7a). Notably, colonies were observed to grow on SD/-Trp-Leu and SD/-Trp-Leu-Ade-His media with the combinations involving *MebHLH149* with either *MePRE5* or *MePRE6*. This growth pattern indicates a potential interaction of MebHLH149 and MePRE5 and MePRE6 proteins (Figure 7b). To further confirm the interaction of MebHLH149 with MePRE5 and MePRE6 in vivo, we cloned MebHLH149 and MePRE5/6 into the pNC-BiFC-Enn and pNC-BiFC-Ecc vectors, respectively, and co-expressed them in *Nicotiana benthamiana* for BiFC detection (Figure 7c). These results substantiate the interaction of the MebHLH149 protein with the PRE5 and PRE6 proteins within *N. benthamiana* plant cells.

## 3. Discussion

Over the last 10 years, CBB has significantly affected the yield of cassava, a critical staple crop. Despite its importance, our understanding of resistance genes associated with this disease is still limited. The present study was designed to investigate the expression patterns of the *MebHLH149* gene in cassava when exposed to *Xpm* CHN01 infection and hormone stress, to shed light on its potential role in the plant’s immune system. The findings indicate that *MebHLH149* is actively involved in the early stages of the plant-pathogen interaction during *Xpm* CHN01 infection, implying a role in the defense mechanism of cassava. Additionally, this study identified *MebHLH149* as an elements responsive to key plant hormones, namely MeJA, SA, and ABA. To further understand the potential signaling pathways influenced by *MebHLH149*, the expression patterns of genes associated with the ABA, MeJA, and SA pathways were examined. The data revealed a notable increase in the expression of genes related to these hormone pathways follow foliar application of the respective hormones. This observation supports the hypothesis that *MebHLH149* plays a role in the signaling cascades mediated by ABA, MeJA, and SA.

Plant hormones are essential in orchestrating the immune responses of plants, creating complex networks that help combat a variety of diseases caused by external factors. These hormones form a protective barrier that not only prevents the progression of disease but bolsters the plant’s overall defense capabilities [24,25]. The significance of SA in plant disease resistance was first reported by White et al., who found that the application of SA to tobacco plants induced symptoms of tobacco mosaic virus (TMV). Intriguingly, these plants also displayed reduced lesion size, with some showing up to a 90% decrease post-infection, highlighting the ability of SA to enhance the resistance of tobacco to TMV [26]. This discovery underscores the importance of this study in clarifying the novel role of the *MebHLH149* gene in the immune response of cassava when treated with plant hormones.

To address the reduction in cassava yield caused by *Xpm*, previous studies employed VIGS as a means to validate the disease-resistance capabilities of various genes. This VIGS technology effectively silences specific target genes within plants, eliminating the need to create transgenic lines or conduct extensive mutant screenings. Its efficiency and simplicity make it particularly advantageous for exploring genes that are involved in plant growth, defense mechanisms, and other critical functions, rendering it an ideal tool for gene function discovery in plant seedlings [27]. For example, prior investigations utilized VIGS to silence the *MeDELLA1*, *MeDELLA2*, *MeDELLA3*, and *MeDELLA4* genes in cassava, aiming to understand their roles. Plants with these genes silenced showed reduced expression levels of genes associated with defense, suggesting a weakened ability to fend off CBB [28]. In a similar vein, a different study scrutinized the expression patterns of three *MeWHYs* genes under various conditions and then applied VIGS to silence them. The silencing of these genes resulted in an upsurge in bacterial populations and a concurrent decline in the disease resistance of cassava [29]. Following this precedent, we applied VIGS to investigate the role of the *MebHLH149* gene in disease resistance. Our findings revealed that once the *MebHLH149* gene was successfully silenced, subsequent inoculation with *Xpm* CHN01 led to more extensive water-soaked lesions and a greater bacteria count when compared to the negative control group. This outcome underscores diminished resistance in cassava following the silencing of the *MebHLH149* gene, thereby highlighting its positive regulatory influence in the plant’s defense mechanisms against CBB.

The resistance of cassava to the CBB pathogen is significantly enhanced by the *MebHLH149* gene, which may function through its combined action with its regulated genes. However, the scarcity of resistance gene germplasm, high genetic heterozygosity, heterozygous polyploidy, and low fertility in cassava present challenges for conventional breeding methods aimed at crop improvement. Given these challenges, progress in conventional breeding is often lengthy and arduous. Consequently, genetic engineering has become an invaluable alternative and complementary approach for enhancing cassava desirable agronomic traits [30]. In the context of this study, we constructed a plant overexpression vector and successfully obtained transgenic seedlings. The ultimate aim was to investigate whether the *MebHLH149* gene confers enhanced resistance to CBB in the overexpressing cassava plants.

This study used the STRING database to predict potential interaction between the MebHLH149 protein and two candidate proteins, MePRE5 and MePRE6. We further explored their respective expression patterns to gain insights into the involvement of these proteins during the cassava–*Xpm* interaction. The findings revealed that the expression levels of both *MePRE5* and *MePRE6* genes were negatively correlated with the silencing of the *MebHLH149* gene; conversely, their expression was positively influenced by the overexpression of *MebHLH149*. This correlation implied potential interaction between MebHLH149 with the MePRE5 and MePRE6 proteins. To confirm these interactions, we employed two complementary experimental techniques, Y2H and BiFC assays. These results highlight the biological significance of the MebHLH149 interaction with MePRE5 and MePRE6.

In conclusion, this study established that the MebHLH149 protein plays a pivotal role in three key plant signaling pathways: ABA, MeJA, and SA. These pathways are indispensable to the plant’s defense mechanisms against various biotic stressors. Notably, *MebHLH149* exerts a positive regulatory impact on the cassava’s resistance to CBB, a devastating condition that can significantly impact crop yields and quality. By enhancing the plant’s ability to respond to pathogens through its involvement in these critical signaling cascades, *MebHLH149* contributes to strengthening the plant’s overall disease resistance and resilience.

## 4. Materials and Methods

### 4.1. Plant Material and Strains

The cassava varieties South China No. 9 (SC9) and ZME642 are cultivated in the National Cassava Germplasm Repository. Situated in Danzhou, Hainan Province, China, with geographical coordinates of 109°30′ E and 19°30′ N, the repository offers an ideal environment for the cultivation and research of this specific cassava variety. *Nicotiana benthamiana* plants were cultivated under controlled environmental conditions within a temperature-controlled chamber at the Institute of Tropical Bioscience and Biotechnology, Chinese Academy of Tropical Agricultural Sciences. The complete genomic data of Chinese *Xpm* strain CHN01 have been deposited to the NCBI GenBank database under accession nos. CP083575, CP083576 (pXPM38), CP083577 (pXPM44), CP083578 (pXPM45), and CP083579 (pXPM48) [7]. This strain was kindly provided by the Key Laboratory of Tropical Biological Resources Sustainable Utilization, Hainan University, Hainan Province, China.

### 4.2. Quantitative Real-Time PCR

Total RNA extraction and reverse transcription experiments from leaves were performed using the RNAprep Pure Plant Plus Kit (TIANGEN, Beijing, China) and the RevertAid First Strand cDNA Synthesis Kit (Thermo Scientific, Waltham, MA, USA) under the manufacturer’s instructions. The reverse-transcribed cDNA template was diluted tenfold for utilization as the template for quantitative real-time PCR and mixed with 2× ChamQ Universal SYBR qPCR Master Mix* (Vazyme, Nanjing, China) and primers for real-time fluorescence quantitative PCR analysis (ABI/QuantStudio 6 Flex, Singapore). Each experiment was carried out with at least 3 biological replicates, and data analysis was executed using the 2^−ΔΔCt^ method. Data analysis employed Dunnett’s test as the post hoc test following ANOVA (analysis of variance). Primer sequences are presented in Appendix A.

### 4.3. Xpm CHN01 and Hormone Stress in Cassava

To investigate the function of the *MebHLH149* gene in the immune response of cassava, leaf samples of the cassava variety SC9 were inoculated with *Xpm* CHN01 and samples were collected at 0 h, 1 h, 3 h, and 6 h, followed by the analysis of *MebHLH149* gene expression patterns using RT-qPCR (using 0 h as the control) [31]. To induce hormonal stress in cassava, ABA, MeJA, and SA were sprayed onto the leaves, samples were collected at 0 h, 3 h, 6 h, 9 h, 12 h, and 24 h to investigate the expression patterns of *MebHLH149* and related genes (using 0 h as the control) [32]. The concentrations of the hormones utilized in these experiments were 100 μmol/L MeJA, 100 μmol/L ABA, and 2 mmol/L SA. Primer sequences employed in the present study are provided in Appendix A.

### 4.4. Identification of MebHLH149

The accession number of the cassava *bHLH149* gene is Manes.09G177600. The coding region (CDS) sequence of the cassava *bHLH149* gene, which is 600 bp in length, was retrieved from the *Manihot esculenta* v6.1 genome in the phytozome v13 (https://phytozome-next.jgi.doe.gov/ (accessed on 24 July 2023)) database. Specific primers were designed using the coding region sequence, and the primers are shown in Appendix A. Meanwhile, the reverse transcription product cDNA of the SC9 cassava leaves was used as the amplification template for PCR amplification. The reaction conditions of PCR amplification are shown in Appendix A.

### 4.5. Transient Gene Expression in N. benthamiana Leaves

Initially, the subcellular localization prediction of the MebHLH149 protein was conducted via the online website WoLF PSORT (https://wolfpsort.hgc.jp/ (accessed on 31 July 2023)); and the nuclear localization signal sequence prediction of the MebHLH149 protein was executed using the online tool NovoPro (https://www.novopro.cn/tools/nls-signal-prediction.html/ (accessed on 20 July 2024)).

The vector utilized for subcellular localization of MebHLH149 protein was pNC-Green-SubN (GFP). The Nimble Cloning method was employed to amplify the target gene fragment, with 20 bp of universal adapter sequences appended to both termini of the gene-specific primers. The upstream primer sequence was 5′-agtggtctctgtccagtcct-3′, and the downstream primer sequence was 5′-ggtctcagcagaccacaagt-3′. Plasmids harboring the pCE2 TA/Blunt-Zero-*MebHLH149* recombinant vector were used as templates for amplification. Gel extraction was carried out for positive bands obtained from amplification. The primers used for amplification are presented in Appendix A. The recovered products were combined with Nimble Mix and an appropriate quantity of pNC-Green-SubN (GFP) plasmid, and the mixture was incubated in a PCR machine at 50 °C for 50 min before being placed on ice for transformation into *Escherichia coli.*

Ultimately, plasmids were extracted from the bacterial liquid with correct sequencing and then transformed into *Agrobacterium* GV3101 Psoup-p19. Suspensions of *A. tumefaciens* harboring the recombinant plasmid pNC-Green-SubN-*MebHLH149* (GFP) and the empty vector pNC-Green-SubN (GFP) were independently injected into the abaxial side of *N. benthamiana* leaves. After one day of dark incubation followed by two days of weak light incubation, fluorescence signals were observed under a confocal laser scanning microscope (LEICA, Wetzlar, Germany).

### 4.6. Virus-Induced Gene Silencing in Cassava

The cassava common mosaic virus (CsCMV) VIGS vector was used to construct the *MebHLH149* silencing vector pCsCMV-*MebHLH149*. The pCsCMV-NC vector utilized in this experiment was developed by Dr. Yan Pu and colleagues at the Chinese Academy of Tropical Agricultural Sciences [33]. First, specific 300 bp fragments of *bHLH149* were selected using the online SGN VIGS Tool (https://vigs.solgenomics.net/ (accessed on 31 July 2023)). Primers were designed based on this sequence for amplification and connection to the target vector (pCsCMV-NC). Nimble Cloning was employed to ligate the target fragment and the pCsCMV-NC vector, ultimately obtaining the pCsCMV-*MebHLH149* recombinant vector required for VIGS. Firstly, the plasmid was transformed into *Agrobacterium tumefaciens*, and the bacterial suspension was subsequently inoculated into the leaves of the cassava variety SC9 that had grown for approximately 40 days. The cassava leaves of the negative control group was inoculated with pCsCMV-NC, and those of the positive control group were inoculated with pCsCMV-*ChlI345*, and the experimental group was inoculated with *Agrobacterium tumefaciens* of pCsCMV-*MebHLH149*.

Subsequently, on the 26th day post-infiltration, the silencing efficiency in test plants was detected by real-time quantitative PCR. *MebHLH149*-silenced plants were inoculated with *Xpm* CHN01 on their newly emerged leaves, and their phenotypic alterations were observed at 0d, 1d, and 2d to analyze gene expression and disease resistance in cassava leaves.

### 4.7. Genetic Transformation of the MebHLH149 Gene

*MebHLH149* gene overexpression was accomplished using the pNC-Cam1304-MCS35S vector. In this experiment, the *MebHLH149* gene was cloned into the pNC-Cam1304-MCS35S vector using the same method as described in Section 4.5. Once the pNC-Cam1304-MCS35S-*MebHLH149* recombinant vector was constructed, it was transformed into *Agrobacterium* strain LBA4404. The *Agrobacterium* containing pNC-Cam1304-MCS35S-*MebHLH149* and the empty vector pNC-CamM1304-MCS35S was cultivated at 28 °C until the OD_600_ value ranged from 0.75 to 1.0. Then, the *MebHLH149* gene was transformed into the cassava friable embryogenic callus (FEC). To view the workflow of the *MebHLH149* gene infecting cassava FEC, please refer to Appendix A. The composition of the medium used for tissue culture and transformation experiments referred to the configuration method of Nyaboga et al. [30].

### 4.8. Screening of MebHLH149 Interacting Protein

To explore the interaction of proteins with MebHLH149, we employed the online String database (https://cn.string-db.org/cgi/input?sessionId=bYYPxcIVb7Mw/ (accessed on 7 December 2023)) to predict a protein–protein interaction network involving the MebHLH149 protein (see Appendix A). We selected high-scoring proteins containing the bHLH domain, MePRE5 and MePRE6, for subsequent validation.

### 4.9. Y2H Validation of Protein–Protein Interactions

The vectors employed in this experiment were pNC-GADT7 (AD) and pNC-GBKT7 (BD). The AD domain is derived from the transcription factor CAL4 and is located in the CAL4 activation domain at the C-terminal. Similarly, the BD domain is also from the transcription factor CAL4 and is located in the CAL4 DNA binding domain at the N-terminal. Firstly, the nucleotide sequence coding for MebHLH149 was cloned into the pNC-GADT7 vector, while the nucleotide sequences coding for MePRE5 and MePRE6 were cloned into the pNC-GBKT7 vector. The amplification and ligation of the gene fragments and subcellular vector construction were performed according to the methods described in Section 4.5.

Prior to performing co-transformation, self-activation assays of the MebHLH149, MePRE5, and MePRE6 proteins were primarily conducted to observe whether there was any self-activation phenomenon. Subsequently, six groups of recombinant plasmids, namely AD + BD, AD-MebHLH149 + BD, BD-PRE5 + AD, BD-PRE5 + AD-MebHLH149, BD-PRE6 + AD, and BD-PRE6 + AD-MebHLH149, were co-transformed into the AH109 yeast competent cells. After the transformation, 20–50 μL were taken, respectively, and spread onto the SD/-Trp-Leu solid medium. When single colonies grew, the single colonies were picked and transferred to the SD/-Trp-Leu liquid medium for activation. The cells were centrifuged and the precipitate was re-suspended and diluted in four gradients according to the gradient dilution method and dropped onto the SD/-Trp-Leu and SD/-Trp-Leu-Ade-His media. The cultures were inverted and incubated at 29 °C.

### 4.10. BiFC Validation of Protein–Protein Interactions

The pNC-BiFC-Enn and pNC-BiFC-Ecc vectors were used in these experiments. To verify whether the MebHLH149 protein interacts with the PRE5 and PRE6 proteins, the MebHLH149 coding sequence was cloned into the pNC-BiFC-Enn vector, while the MePRE5 and MePRE6 coding sequences were cloned into the pNC-BiFC-Ecc vector. The constructed vectors were transformed separately into *A. tumefaciens* GV3101 Psoup-p19. A bacterial suspension was prepared using infiltration suspension (10 mM MES, 10 mM MgCl_2_, 100 µM acetosyringone) and adjusted to OD_600_ = 0.5. Equal volumes of the mixed bacterial suspension were prepared and left to stand in the dark for approximately 2 h. Using sterile disposable syringes, the mixed bacterial suspensions were injected separately onto the underside of *N. benthamiana* leaves. At 3 dpi, after staining with DAPI, the YFP signal was detected using confocal laser scanning microscopy (TCS SP8, Leica), and the in vivo interactions of MebHLH149, MePRE5, and MePRE6 were analyzed.

### 4.11. Statistical Analysis

All bar charts in this study were created based on data from three biological replicates, and normality tests were performed using Prism 8.0.2 software (https://www.graphpad.com/ (accessed on 29 April 2023)). Significant differences were assessed using ANOVA and Dunnett’s test.

### 4.12. Accession Numbers

The sequence data used in this study can be found in the Phytozome v13 and NCBI (https://www.ncbi.nlm.nih.gov/ (accessed on 25 July 2023)) online databases. The accession numbers for the *MebHLH149* gene and related research genes are presented in Appendix A.

## Figures and Tables

**Figure 1 plants-13-02422-f001:**
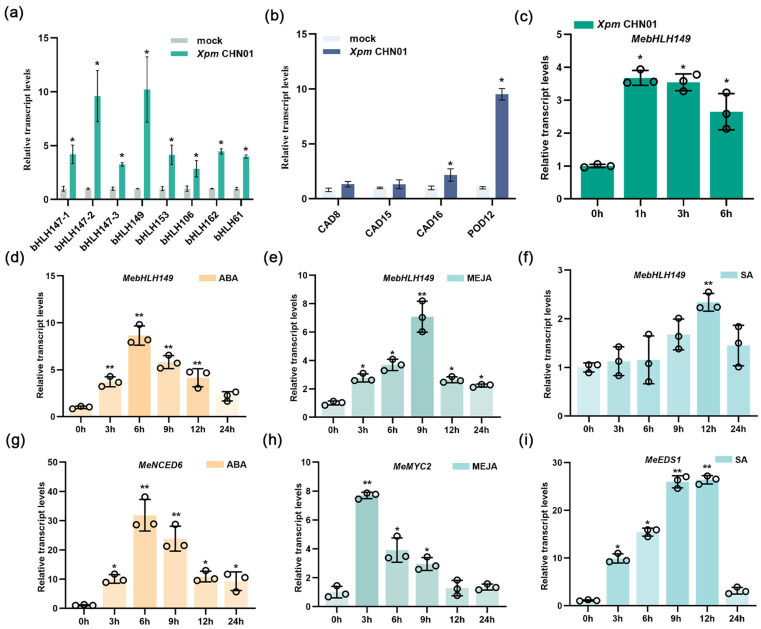
Expression patterns of the *MebHLH149* gene in response to various stressors. (**a**) Response of *MebHLHs* to *Xanthomonas phaseoli* pv. *Manihotis* (*Xpm*) infection. (**b**) Co-expression analysis with lignin and ROS-related genes. Samples were collected from cassava (*Manihot esculenta* Crantz) leaves 10 days after inoculation with *Xanthomonas phaseoli* pv. *manihotis* strain CHN01 (*Xpm* CHN01) when the plants were around 40 days old. (**c**) Early response dynamics of *MebHLH149* to *Xpm* CHN01 inoculation; (**d**–**f**) *MebHLH149* expression under hormonal stress induced by ABA, MEJA, and SA; (**g**–**i**) Expression analysis of hormone-associated genes. All experiments were performed with three independent biological replicates to ensure the accuracy and consistency of the results. The 2^−ΔΔCt^ method was used for quantitative analysis of gene expression, with statistically significant differences indicated as follows: * *p* < 0.05 and ** *p* < 0.01, denoting the level. Significant differences were assessed using ANOVA and Dunnett’s test.

**Figure 2 plants-13-02422-f002:**
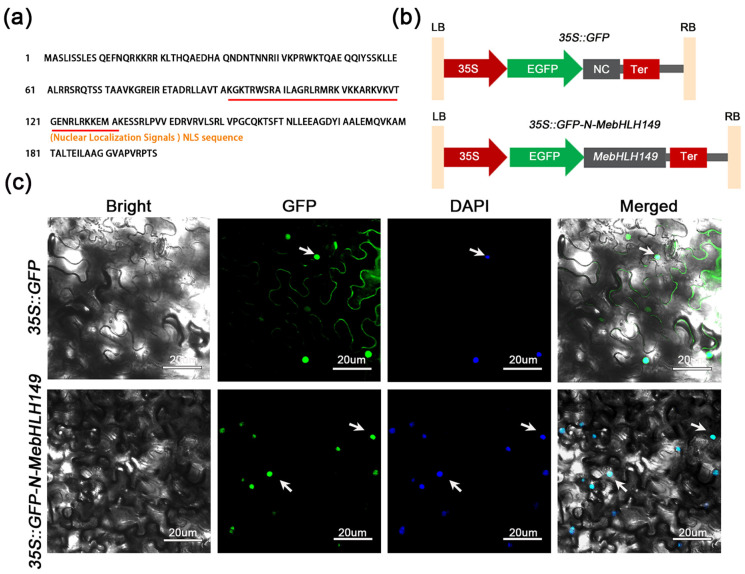
Subcellular localization of MebHLH149 protein: (**a**) the nuclear localization signal sequence of MebHLH149 protein. The red line represents the nuclear localization signal (NLS) sequence; (**b**) schematic representation of the subcellular localization expression construct. LB: left border; RB: right border; 35S: 35S promoter; EGFP: enhanced green fluorescent protein; NC: Nimble Cloning; Ter: NOS terminator; (**c**) subcellular localization of MebHLH149 protein. Tobacco leaves were infiltrated with either an empty vector or a recombinant plasmid containing MebHLH149 protein, both fused with the GV3101 Psoup-p19 vector, over a period of 3 days. GFP signals within the infiltrated areas were visualized using a confocal laser scanning microscope. To enhance the clarity of the cell nuclei, 1 mg/mL 40,6-diamidino-2-phenylindole (DAPI) was also infiltrated. The arrows represent the colocalization of GFP and DAPI staining in the nuclei of *Nicotiana benthamiana* leaves. Scale bar = 20 μm.

**Figure 3 plants-13-02422-f003:**
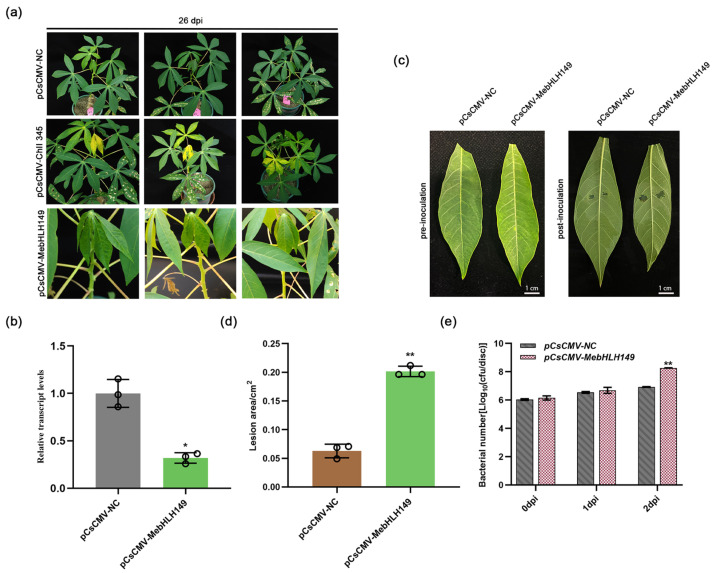
Silencing of *MebHLH149* gene: (**a**) phenotypes of *MebHLH149*-silenced cassava plants captured on the 26th day after *MebHLH149* gene silencing; (**b**) detection of *MebHLH149* gene silencing efficiency; (**c**) phenotypes of cassava plants after *MebHLH149* gene silencing and inoculation with *Xpm* CHN01; (**d**) statistical analysis of lesion area; (**e**) statistical analysis of bacterial count. pCsCMV-*ChlI345* represents the positive control; pCsCMV-NC represents the negative control; pCsCMV-*MebHLH149* represents the experimental group. All experiments were conducted with three independent biological replicates, and the analysis was performed using the 2^−ΔΔCt^ method. Significant differences are indicated as follows: * *p* < 0.05 and ** *p* < 0.01. Significant differences were assessed using ANOVA and Dunnett’s test.

**Figure 4 plants-13-02422-f004:**
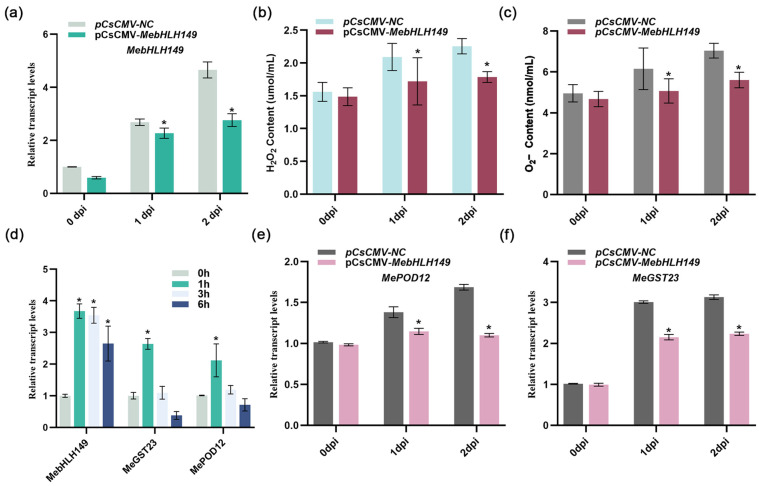
Role of *MebHLH149* in conferring cassava resistance to CBB: (**a**) impact of *MebHLH149* gene silencing on the expression of key immune response genes in cassava assessed following *Xpm* CHN01 inoculation; (**b**) effect of *MebHLH149* on H_2_O_2_ production; (**c**) effect of *MebHLH149* on O_2_^−^ production; (**d**) expression dynamics of *MebHLH149* and two other ROS-associated genes, *MePOD12* and *MeGST23*, in response to *Xpm* CHN01 infection; (**e**,**f**) relative expression levels of *MePOD12* and *MeGST23* genes following *MebHLH149* silencing and subsequent *Xpm* CHN01 inoculation. Each experiment was performed with three biological replicates. pCsCMV-NC represents the negative control; pCsCMV-*MebHLH149* represents the experimental group. Data analysis was conducted using the comparative 2^−ΔΔCt^ method to quantify relative gene expression. Statistically significant differences indicated as * *p* < 0.05 denote the efficacy of *MebHLH149* silencing on the plant’s defense response and ROS production against CBB. Significant differences were assessed using ANOVA and Dunnett’s test.

**Figure 5 plants-13-02422-f005:**
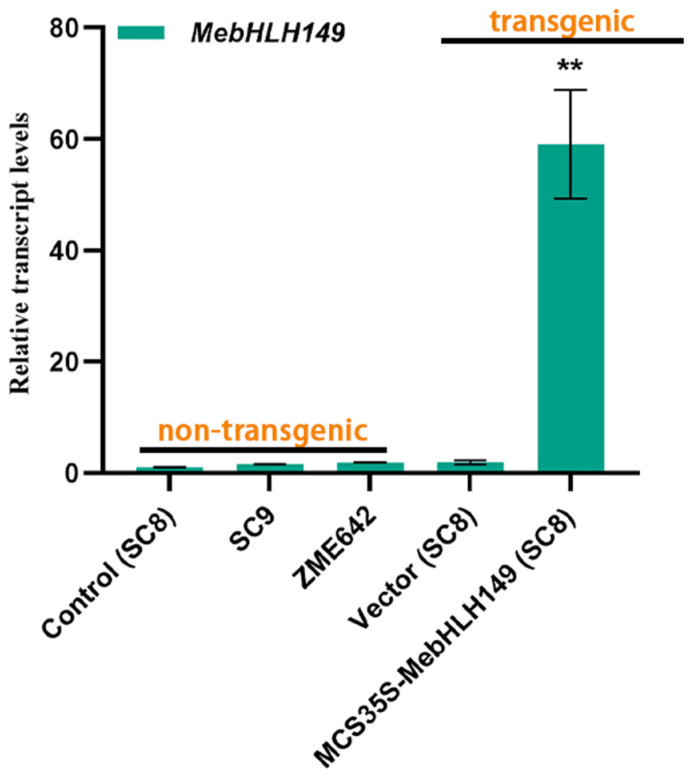
The expression levels of *MebHLH149* in non-transgenic and transgenic cassava were investigated: Control (SC8): non-transgenic cassava varieties SC8; SC9 and ZME642: non-transgenic cassava varieties; Vector (SC8): the transgenic cassava was obtained through the genetic transformation of the overexpressed empty vector in the brittle callus of the SC8 cassava variety; MCS35S-*MebHLH149* (SC8): *MebHLH149* was cloned into the overexpression vector and transformed into the brittle callus of the SC8 cassava variety. The transgenic cassava was obtained through genetic transformation. Each experiment was performed with three biological replicates, and data were analyzed using the 2^−ΔΔCt^ method. Statistical significance was determined at ** *p* < 0.01. Significant differences were assessed using ANOVA and Dunnett’s test.

**Figure 6 plants-13-02422-f006:**
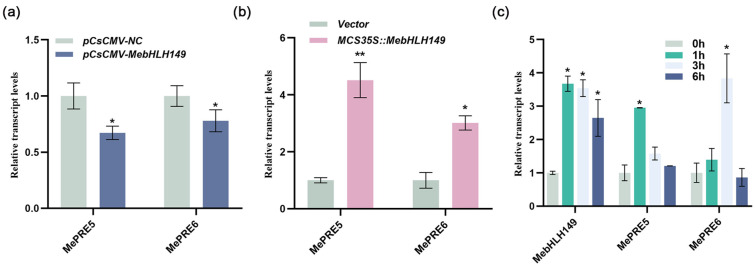
Expression patterns of *MePRE5* and *MePRE6* genes in cassava plants: (**a**) relative expression patterns of *MePRE5* and *MePRE6* genes following the silencing of *MebHLH149* gene; (**b**) relative expression levels of *MePRE5* and *MePRE6* genes after overexpression of *MebHLH149* gene; (**c**) early expression levels of *MePRE5* and *MePRE6* genes under *Xpm* CHN01 infection. pCsCMV-NC: the negative control in the virus-induced gene silencing experiment; pCsCMV-*MebHLH149*: the experimental group with the transient transformation of the *MebHLH149* gene in the virus-induced gene silencing experiment; Vector: the transgenic cassava was obtained through genetic transformation of the overexpressed empty vector in the brittle callus of the SC8 cassava variety; MCS35S-*MebHLH149*: *MebHLH149* was cloned into the overexpression vector and transformed into the brittle callus of the SC8 cassava variety. Each experiment was performed with three biological replicates, and data were analyzed using the 2^−ΔΔCt^ method. Statistical significance was determined at * *p* < 0.05 and ** *p* < 0.01. Significant differences were assessed using ANOVA and Dunnett’s test.

**Figure 7 plants-13-02422-f007:**
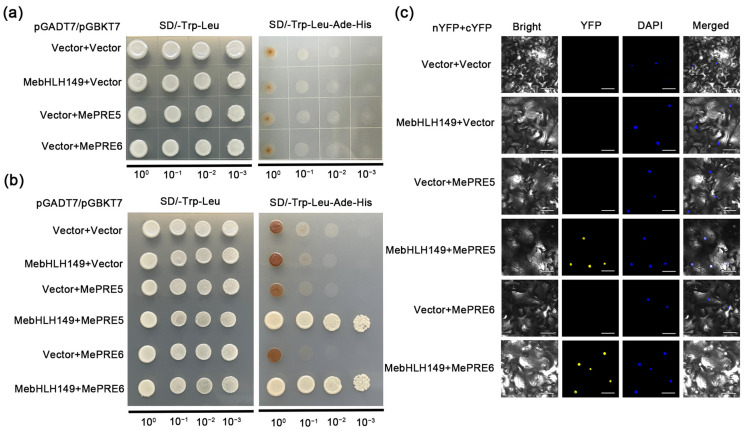
MebHLH149 physically interact with MePRE5 and MePRE6: (**a**) the results of the self-activation verification indicated that neither MebHLH149 nor the MePRE5 and MePRE6 proteins showed self-activation; (**b**) the yeast two-hybrid assay showed the interaction between MebHLH149 and MePRE5 as well as MePRE6 in yeast; (**c**) the BiFC detection showed the in vivo interaction between MebHLH149 and MePRE5 as well as MePRE6. Bar = 20 μm.

## Data Availability

The data that support the findings of this study are available from the corresponding author upon reasonable request.

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
