# Peer review of "Expression Pattern and Functional Analysis of MebHLH149 Gene in Response to Cassava Bacterial Blight"

_plants, 2024, doi:10.3390/plants13172422_

Round 1
Reviewer 1 Report
Comments and Suggestions for Authors
Manuscript plants-2993263 describes results of an investigation on the role of the MebHLH149 gene during the interaction of cassava plants with Xanthomonas phaseoli pv. manihotis. The authors report the association of the MebHLH149 function with resistance to pathogen infection, production of ROS, its responsiveness to plant hormones, as well as the subcellular localization of the MebHLH149 protein and its physical interaction with other helix-loop-helix proteins that are predicted to be part of the MebHLH149-associated protein-protein interactome.
GENERAL COMMENTS:
_ The manuscript presents very interesting and relevant information for the scientific community. The experimental approaches utilized are very sound, and the figures are, for the most part, well presented.
_ The study was conducted with a crop plant, important for human consumption in developing countries, which makes the story more interesting.
_ Unfortunately, the current version of the manuscript is of very poor quality. The English language needs extensive revision in order to get it ready for publication. I regret to say that, despite the manuscript being subjected to professional proofreading (the authors provide evidence for that fact) the quality of the English language is still very deficient, specially in the materials and methods and supplementary information sections. The meaning of many sentences, and even some long sections, is very difficult to understand.
_ The manuscript also requires a better organization of its structure, mainly of the results section. Long stretches of information included in this section are mere technical aspects related to the materials and methods. This latter section is very poorly written, mainly sub-section 4.8, which is a lab protocol, not acceptable for publication in any scientific journal.
SPECIFIC COMMENTS:
_ There are way too many specific comments to be listed in this review. Since I believe the story is interesting and relevant for the scientific community, I have carefully annotated the PDF proof aiming to provide some suggestions that can help improve the quality of the manuscript. However, the authors still need to subject the manuscript to proofreading by an idoneous native English speaker before submitting the new version for publication.

The English language requires extensive revision.
Reviewer 2 Report
Comments and Suggestions for Authors
This research opens the door on the role(s) of HLH 149 protein, in plant immunity beyond other studies that describe the role in more plant growth, physiology, interactions. The manuscript is well written and presented research. The researchers provide evidence for the interaction and potential functional role of HLH149, in cassava and make comparisons to the Arabidopsis model system. The research is elucidating an important problem that threatens food security for countries that grow Cassava, one of the top 12 food crops in the world. The research is at the beginning of a long process to identify the critical protein interactions in Cassava immunity that might be used to increase tolerance against bacterial pathogens causing CBB. The Methods and statistical analyses provide adequate instruction and apply to results as presented. Figures and images all nicely displayed, and the references cited appear to provide further support of statements. Doi addresses included in references will be greatly appreciated by readers.
Line 126, In Results --...'thereby play a pivotal role in the immune defense mechanisms of cassava.'-- Discussion statement.
Lines 156-161, in Results-- are Discussion statements.
Lines 198-200, in Results -- is Discussion point.
Lines 293-295 in Results are Discussion.
Line 354, need to correct '1-hor' to '1-hr'
Line 559 need to italicize 'E. coli',
Line 249 in Ref 16, 'need spaces- TheArabidopsisbHLH
Formatting: Scientific names should be Italicized in Reference List, which is the normal format for scientific publications.
Reviewer 3 Report
Comments and Suggestions for Authors
The current manuscript by Cui et al. has analyzed the role of the transcription factor MebHLH149 in cassava bacterial blight disease. The authors have done a good job of experimental designing, performing experiments, and data analysis. Overall, most conclusions drawn from the data are satisfactory, however, the following concerns must be addressed to improve the manuscript before it can be accepted for publication.
The manuscript needs substantial editing for English language, grammatical, and syntax errors as well as for scientific writing. I suggest the authors take the help of a professional English language editing service to address this issue.
Graphical abstract: The graphical abstract is non-informative. It appears just a collection of random symbols of DNA here and there, and text connected by arrows without giving any specific scientific insights about gene regulation. Please prepare a relevant graphical abstract giving an appropriate take-home message arising from this work.
Results, Line 196: In Figure 2b, the colocalization of green and blue signals appears to happen in both empty vector as well as GFP-tagged MebHLH149. Therefore, this result is not convincing.
Results, Line 99 “The gene sequence under investigation” – Please include the rationale for selecting the genes in question and a table of selected genes with their respective gene ID.
Title of the manuscript, line 2 – “MebHLH149 2 genes” in the title indicates the authors have studied multiple genes, however, in the abstract the authors write – “bHLH149 transcription factor” (line 11) suggesting that the current work focuses on one gene. Please correct the discrepancies.
Abstract: Lines 18-21 – Rephrase the sentence. It does not read well.
Line 91: “Cassava plants will be inoculated with Xpm CHN01”, Line 94: Additionally, the variation in cassava resistance under silenced conditions of the MebHLH149 gene will be explored using VIGS technology – Please correct the English language and grammatical errors. The sentences are written in the future tense.
Change ‘CDNA’ to ‘cDNA’ everywhere in the manuscript.
Please change ‘Xpm CHN01 stress’ to ‘Xpm CHN01 infection’.
Comments on the Quality of English LanguageModerate editing is required for the English language and better readability.
Reviewer 4 Report
Comments and Suggestions for Authors
There are many unreasonable data analyses in this manuscript, which weakens the credibility of the data. The writing is not smooth enough, and the logic is confusing in some place.
1. Which gene does the ck in Figure 1(a) (b) refer to? Each gene needs its own ck.
2. Figure 1. (a)” MebHLH149 expression in relation to Xpm CHN01 inoculation”, This description is not accurate enough, because more than one gene was measured here.
3. Figure 2, the figure legend is incomplete, and the meaning of the abbreviations is not introduced. For example, what is NC? Why is the GFP of the empty vector transformant also localized to the cell nucleus? The text does not introduce relevant specific information. It is also necessary to add a description to explain the meaning of the arrows.
4. Figure 3, RNAi usually has poor specificity. This manuscript only selects one site for interference and only selects one interference line for analysis, which is far from enough. It is impossible to confirm whether the various phenotypes observed are causally related to the decreased expression of the target gene.
5. Figure 5 (b) is puzzling. How does this data prove overexpression of the MebHLH149?
Reviewer 5 Report
Comments and Suggestions for Authors
The main conclusion of this manuscript is that MebHLH149 may be a positive regulator of cassava's resistance to CBB. However, the data supporting this conclusion are superficial, and the original data are insufficient to confirm the core argument. It is recommended to redesign the experiments and focus on key aspects to ensure that the framework is more rigorous and coherent in logic.
Key issues include:
1. MebHLH149 may affect ROS accumulation, but the specific mechanism remains underexplored. The reduction in ROS levels and MePOD12 and MeGST23 expression after silencing MebHLH149 suggests that MebHLH149 may regulate ROS accumulation by affecting the expression of these antioxidant enzyme genes. This conclusion must be further verified by additional experiments, such as overexpression studies, direct binding assays, and protein interaction analysis to confirm the specific role of MebHLH149 in ROS metabolism.
2. The MebHLH149 gene responds to plant hormones ABA, MeJA, and SA, suggesting that it may be involved in the signaling pathways mediated by these hormones in immunity. However, further evidence is needed. Besides, MePRE family proteins are always involved in the GA signaling pathway, which is important for plant growth and development (rather than immunity). Confusingly, most of the experiments were conducted to prove the existence of the interactions between MebHLH149 and MePRE5/6 proteins.
3. In terms of manuscript writing, a more rigorous and professional approach is encouraged. Authors claimed that "verification of whether transgenic plants overexpressing MebHLH149 are resistant to CBB at later growth stages" in Section 2.9, but no data support this point, and the conclusion at the end of this section is irrelevant; Sections 2.4, 2.5, and 2.6 are not suitable to present in the main text.
Comments on the Quality of English LanguageNeed to be polished.
Round 2
Reviewer 4 Report
Comments and Suggestions for Authors
Figure 2, since the authors believe that the GFP of the empty vector transformant can localize to the cell nucleus, then the nuclear localization of MebHLH149 might be the result of the empty vector's function rather than the true localization of MebHLH149. How do the authors exclude this possibility?
Figure 5 (b) only shows that the genetic transformation of the MebHLH149 gene in cassava was successful. The evidence of overexpression is in Figure 6 (b), please reorganize the data.
Comments on the Quality of English LanguageThere are a large number of irregularities in the writing and revision of the paper, which requires careful revision throughout.
Reviewer 5 Report
Comments and Suggestions for Authors
It is a pity that the authors could not explore further due to limited conditions. In the current context, the experimental results are presented clearly, in line with the restrained expression of "expression and functional analysis" in the title. In terms of the main text, I think the "Results" section should emphasize the key points. Sections such as "Cloning of cassava bHLH149 gene" and "Silencing of cassava MebHLH149 gene using pCsCMV-NC vector" should be moved to the experimental methods section for greater clarity.
Round 3
Reviewer 4 Report
Comments and Suggestions for Authors
The authors have carefully modified all the issues raised.